# Inorganic Nitrogen Transport and Assimilation in Pea (*Pisum sativum*)

**DOI:** 10.3390/genes13010158

**Published:** 2022-01-17

**Authors:** Benguo Gu, Yi Chen, Fang Xie, Jeremy D. Murray, Anthony J. Miller

**Affiliations:** 1Biochemistry & Metabolism Department, John Innes Centre, Norwich Research Park, Norwich NR4 7UH, UK; Benguo.gu@jic.ac.uk (B.G.); yi.chen@jic.ac.uk (Y.C.); 2CAS-JIC Centre of Excellence for Plant and Microbial Science (CEPAMS), Shanghai Institutes for Biological Sciences, Chinese Academy of Sciences (CAS), Shanghai 200032, China; jmurray@cemps.ac.cn; 3National Key Laboratory of Plant Molecular Genetics, CAS Center for Excellence in Molecular Plant Sciences, Chinese Academy of Sciences (CAS), Shanghai 200032, China; fxie@cemps.ac.cn

**Keywords:** pea, *Pisum sativum*, inorganic nitrogen, transport, assimilation

## Abstract

The genome sequences of several legume species are now available allowing the comparison of the nitrogen (N) transporter inventories with non-legume species. A survey of the genes encoding inorganic N transporters and the sensing and assimilatory families in pea, revealed similar numbers of genes encoding the primary N assimilatory enzymes to those in other types of plants. Interestingly, we find that pea and *Medicago truncatula* have fewer members of the NRT2 nitrate transporter family. We suggest that this difference may result from a decreased dependency on soil nitrate acquisition, as legumes have the capacity to derive N from a symbiotic relationship with diazotrophs. Comparison with *M. truncatula*, indicates that only one of three NRT2s in pea is likely to be functional, possibly indicating less N uptake before nodule formation and N-fixation starts. Pea seeds are large, containing generous amounts of N-rich storage proteins providing a reserve that helps seedling establishment and this may also explain why fewer high affinity nitrate transporters are required. The capacity for nitrate accumulation in the vacuole is another component of assimilation, as it can provide a storage reservoir that supplies the plant when soil N is depleted. Comparing published pea tissue nitrate concentrations with other plants, we find that there is less accumulation of nitrate, even in non-nodulated plants, and that suggests a lower capacity for vacuolar storage. The long-distance transported form of organic N in the phloem is known to be specialized in legumes, with increased amounts of organic N molecules transported, like ureides, allantoin, asparagine and amides in pea. We suggest that, in general, the lower tissue and phloem nitrate levels compared with non-legumes may also result in less requirement for high affinity nitrate transporters. The pattern of N transporter and assimilatory enzyme distribution in pea is discussed and compared with non-legumes with the aim of identifying future breeding targets.

## 1. Introduction

The main forms of soil inorganic nitrogen (N) available to pea (*Pisum sativum,* Ps) roots are either nitrate (NO_3_^−^) or ammonium (NH_4_^+^). NO_3_^−^ and NH_4_^+^ transporter families are well described in model species like *Arabidopsis thaliana* (At), but in general there is less information available for legume transporters. Legumes also have the capacity to acquire N from the air through a symbiotic relationship with diazotrophic bacteria, forming specialized nodule structures to host this partnership. As the symbiosis is energetically expensive and the plant supplies the bacteria with a carbon source, the nodule N fixing activity is usually restricted to environmental conditions when soil N supply is limited. For pea, this alternative N supply may result in some differences in the gene families encoding inorganic N transporters and assimilatory enzymes and this is the subject of this review. Pea is particularly interesting when compared with the model legumes *M. truncatula* (Mt) and *Lotus japonicus* (Lj), because the species has been domesticated and bred for agricultural cropping. Furthermore, a reference pea genome has been published that provides the information needed to identify gene families [1]. 

Nitrate transporters can be split into three families, the nitrate transporter 1 (NRT1), NRT2 and CHLORIDE CHANNEL (CLC) families [2]. In *A. thaliana*, examples are found of each type, and these are variously located in the plasma membrane, tonoplast and other endomembrane systems. Ammonium transporters (AMT) are divided into two types, AMT1 and AMT2 and these too can be in different types of membrane. As the utilization of N depends on the primary assimilatory enzymes, we have also taken this opportunity to compare these proteins in pea with other types of plants. Finally, we have considered the regulation and signalling component of inorganic N use in pea [3]. 

In Figure 1, the right (white) half describes N taken from the soil and the left (pink) half is N-fixed in nodules. For the right half, NO_3_^−^ is acquired by roots and then transported to the whole plant by transporters and through the phloem and xylem. This process is regulated by nitrate sensing, signalling and assimilation. After sensing NO_3_^−^ by transceptors, NRT1.1s, a calcium burst mediated by CYCLIC NUCLEOTIDE-GATED CHANNEL 15 (CNGC15) activates CALCINEURIN B-LIKE PROTEIN INTERACTING PROTEIN KINASE IIIs (CIPKIIIs). NODULE INCEPTION (NIN)-LIKE PROTEIN 4 (PsNLP4) is phosphorylated by CIPKIIIs and shuttled into the nucleus to initiate transcription of assimilatory proteins, such as NITRATE REDUCTASE (NIR), NITRITE REDUCTASE (NIA) and the transporter PsNRT2.3. NO_3_^−^ is reduced to NH_4_^+^ by NIA and NIR in the cytosol and plastids. In the left half, gaseous N_2_ in the air can be fixed in nodules by symbiotic diazotrophic bacteria to form NH_4_^+^ which is transported by AMT1s. NH_4_^+^ from multiple origins is assimilated into an organic form as glutamine, by GLUTAMINE SYNTHETASES (GS) and GLUTAMINE OXOGLUTARATE AMINOTRANSFERASE (GOGAT) and then transferred into other organic molecules such as ureides, amides and amino acids to be translocated to the whole plant. Note that in legumes, the main organic N phloem-transported compounds for pea are amides, while in soybean, ureides dominate [4].

## 2. Transporter Families

### 2.1. NRT1.1/NPFs including Transceptors

There are 69 NRT1/ PTR FAMILY (NPF) proteins in pea, which can be divided into eight groups like those identified in *A. thaliana* (Appendix A). The expansion of NPFs, many of which are demonstrated nitrate transporters [6], in both dicotyledon and monocotyledon plants indicates a complexity of NO_3_^−^ sensing and signalling. In *A. thaliana*, AtNPF6.3, also named AtNRT1.1 or CHL1, was the first identified nitrate transporter; it was later designated as a plant transceptor that can sense the concentration of NO_3_^−^ in soil to induce expression of genes for NO_3_^−^ uptake and assimilation, which is called primary nitrate response (PNR). AtNRT1.1 is a dual affinity nitrate transporter and transports NO_3_^−^ across a broad range of external NO_3_^−^, and whose transport activity is reduced by phosphorylation of residue threonine (T) 101. The mutant *chl1-9* has lost high affinity nitrate transport, but is still functional in PNR signalling, indicating that NRT1.1′s function as a nitrate transceptor [7]. In maize, there are two NRT1.1 homologues, ZmNPF6.4 and ZmNPF6.6. ZmNPF6.6 is a high affinity nitrate transporter, while ZmNPF6.4 is a low affinity nitrate transporter that shows preference for transport of chloride [8]. The anion substrate preference of these transporters depends on one amino acid (corresponding to H356 in AtNPF6.3), where a change from tyrosine (Y) to histidine (H) to in ZmNPF6.4 at the NO_3_^−^ binding site can result in NO_3_^−^ selectivity. In the model legume, *M. truncatula*, there are three NRT1.1 orthologues, named MtNRT1.1a, MtNRT1.1b and MtNRT1.1c [5,9] (Appendix A). The substrate preference of these three proteins varies, MtNRT1.1B has the H corresponding to AtNPF6.3:H356, and is NO_3_^−^ selective, MtNRT1.1A has Y at this position and is chloride selective, while MtNRT1.1C has a glutamine (Q) in this position and its substrate(s) is unknown (Appendix A). This suggests that MtNRT1.1B has a similar function to AtNPF6.3. In the pea genome, four close homologs of AtNPF6.3 are found. Three appear to be orthologues of MtNRT1.1a/b/c, while another additional allele of NRT1.1, named PsNRT1.1d, was also found. PsNRT1.1d is a B-type protein with H at the substrate binding site, suggesting it may be NO_3_^−^ selective, but it is phylogenetically closer to the chloride selective A-type proteins MtNRT1.1A (Appendix A). Genomic sequence analysis reveals PsNRT1.1d has four introns, which is the same as A-type NRT1.1s, while B-type NRT1.1s have three introns. PsNRT1.1d is a mix of A and B-type NRT1.1 protein, implying a more complex NO_3_^−^ sensing in pea. 

Recently, in *A. thaliana*, another plant nitrate transceptor was identified. AtNPF4.4/AtNRT1.13, was reported to regulate shoot architecture and flowering time under low NO_3_^−^ conditions. AtNRT1.13, which is not a functional nitrate transporter, acquires nitrate transport activity by a single amino acid substitution (S487P), and the authors suggested that it may physically bind and respond to the nitrate anion [10]. Interestingly, AtNRT1.13 contains a T at the residue corresponding to AtNRT1.1:T101, and this residue was found to be present in virtually all nitrate transporting NPFs [9]. This observation suggests that some NPFs may function as nitrate receptors to sense and regulate NO_3_^−^ signalling with no transporter activity and the transporter structure is only used as a conserved protein posture to bind nitrate anion. In pea, PsNPF4.4 is localised in the same subclade with AtNPF4.3 and AtNPF4.4, implying its possible function in shoot architecture and flowering time (see Appendix A). The next steps in NO_3_^−^ signalling are the processes downstream from the transceptor that yield a calcium signal and transduction including channels and kinases. These are described below in Section 4.

Within the legumes, the long-distance transport of N in the phloem is in organic N forms such as ureides, allantoin, and asparagine, and we suggest less NO_3_^−^ is present than in non-legumes resulting in less phloem NO_3_^−^ transport. In pea, long-distance phloem transport of organic N occurs mainly as amides with also less NO_3_^−^ transport. In *A. thaliana*, the NPF2.13/NRT1.7, NPF2.9/NRT1.9, NPF1.2/NRT1.11 and NPF1.1/NRT1.12 transporters are all important for phloem NO_3_^−^ transport [11,12,13]. Orthologues of these phloem transporters do exist in pea (see Appendix A).

### 2.2. NRT2 and NAR2 Families

NRT2 is a family of high affinity nitrate transporters, which is downstream of the transceptors and the transcription factors, and takes a major role in NO_3_^−^ uptake from soil [14,15]. *A. thaliana* has seven *NRT2* genes, named *AtNRT2.1 to AtNRT2.7*. AtNRT2.1, AtNRT2.2 and AtNRT2.4 are important for the uptake of NO_3_^−^ from soil, while AtNRT2.7 is also functional in NO_3_^−^ storage in the seed [16]. Surprisingly, we found only one full-length NRT2 in the pea genome and named it PsNRT2.3 (Psat4g113000), as it shows the greatest similarity to MtNRT2.3 (see Appendix A). Another feature of NRT2 is that it has alternative splicing forms with varied function [17], but we found only one transcriptional form of PsNRT2.3 from the current pea genome annotation. By using *MtNRT2.1* and *2.2* genomic DNA, we identified two more NRT2 proteins, PsNRT2.1 (Psat4g155600.1) and 2.2 (Psat7g149120.1), in the pea genome. Pea would then have three NRT2 proteins, like *M. truncatula*, but PsNRT2.1 and 2.2 are both shortened and predicted to have only three transmembrane domains, while most NRT2 proteins have at least eight. This indicates possible loss of nitrate transport function in PsNRT2.1 and 2.2. However, the functional activity of PsNRT2.1 and 2.2 still needs to be tested by experiment. The decreased number of NRT2 genes in legumes indicates less reliance on NO_3_^−^, which is possibly balanced by the supply from N-fixation by nodules. Legume seeds contain large amounts of N, with seeds of soybean (*Glycine max*) and *Vigna* sp. containing from 45% to 70% of the total plant N at physiological maturity [18], mainly in the form of storage proteins. The absence in pea and *M. truncatula* of a clear counterpart of AtNRT2.7, which determines NO_3_^−^ content in *A. thaliana* seeds [16], may suggest decreased importance of NO_3_^−^ as an N storage form in legume seeds. This idea is worthy of further investigation.

High-affinity nitrate transporter-activating protein 2 (NAR2)/NRT3, is a partner of NRT2s that facilitates NRT2 localization in the plasma membrane [19,20]. Only one PsNRT3.1 (Psat4g061680) gene was found in the pea genome (see Appendix A), indicating the two component NRT2-NAR2 system is present in pea. We can therefore predict that PsNRT2s and PsNAR2 are co-expressed in pea. In *A. thaliana* and some legume species (such as *M. truncatula* and *L. japonicus*), the *NRT2.1* and *NRT2.2* family members are found adjacent on the chromosome, but in pea the three NRT2 genes are unlinked. This may suggest that, within the *NRT2*s, separate multiple gene duplication events in non-legumes and legumes occurred after the legumes evolved. In many plant species, the NRT2s have an important role in the acquisition of NO_3_^−^ from soil, therefore it seems likely the regulation of their transport function must be closely linked to nodule N fixation activity. 

### 2.3. CLC Family

Large amounts of NO_3_^−^ can be stored in vacuoles and the CLC family transporters together with NRT2s determine the sub-cellular distribution of NO_3_^−^ [16,21,22,23]. There are seven CLC proteins in the *A. thaliana*, named AtCLC-A to G, which localize to various cellular membranes [24]. In pea, we identified eight CLC proteins, named PsCLC-A to H (see Appendix A). A chloroplast localization signal was found only in PsCLC-E, indicating its possible conserved function with AtCLC-E. The sub-cellular localizations of the other seven proteins still need to be tested by experiment. 

As some NRT2s and CLCs are involved in vacuolar accumulation of NO_3_^−^ we have checked published values for tissue NO_3_^−^ concentrations to compare pea with non-legume species grown with the same level of N supply. A few papers have made a direct comparison between pea and a cereal for plants grown under the same N supply (e.g., Gloser et al. 2020) [25]. We have plotted the data from a selection of papers and the tissue NO_3_^−^ concentrations in pea are always up to an order of magnitude lower than non-legume species (see Figure 2). This is true for shoots and seems to be independent of nodulation in pea. 

Vacuolar accumulation of NO_3_^−^ dominates the volume of mature plant tissues, and hence we assume that vacuolar NO_3_^−^ storage is decreased in pea. The lower tissue NO_3_^−^ concentrations may result from less NRT2 activity in pea, as the numbers of CLCs found in pea and *A. thaliana* are very similar. Another factor that might be limiting vacuolar storage of NO_3_^−^ in pea is the activity of the tonoplast proton pumps that energize the transport systems in this membrane. 

### 2.4. SLAC/SLAH Family

SLOW ANION CHANNEL-ASSOCIATED 1 (SLAC1) and SLAC1 HOMOLOGUE (SLAH) family were originally characterised as anion channels regulating stomatal closure [32,33,34], then also confirmed for NO_3_^−^ and chloride loading of the root xylem [32]. Searching the pea genome, we found one PsSLAC1 and four PsSLAHs covering all three subclades of SLAC1, SLAH2/3 and SLAH1/4 (Appendix A), indicating functional conservation of the SLAC/SLAH family proteins in pea.

### 2.5. AMT1 and AMT2 Families

Ammonium can be taken up from soil by ammonium transporters (AMTs). In *A. thaliana*, the AMT1 family includes five proteins and takes the major role in NH_4_^+^ uptake from the soil [35,36], while the single AMT2 functions mainly in root-shoot transport of NH_4_^+^ [37]. There are four AMT1 and five AMT2 family transporters in pea (see Appendix A). The expansion of the AMT2 family, which is also found in crops like rice, implies new functions and a complex regulation of NH_4_^+^ in plants [36]. The increased number of AMT2s probably results from the increased availability of NH_4_^+^ in the flooded soils used for the cultivation of rice and gaseous N fixation in legumes. 

The AMT1s show post-translational regulation with negative feedback through a phosphorylation site near the C-terminus of the protein which is highly conserved [38]. AMT1;1 exists in active and inactive states, probably controlled by the spatial positioning of the C-terminus. NH_4_^+^ triggers the phosphorylation of a conserved threonine residue (T460) in the C-terminus of AMT1;1 in a concentration- and time- dependent way [38]. The T460 phosphorylation level correlates with a decrease of root NH_4_^+^ uptake. The T460 sites are conserved in most *A. thaliana* AMT1s but substituted by an alanine (A) in AtAMT1.5. However, in legumes a common variation is serine (S) at this residue, occurring in *M. truncatula* and pea (see Appendix A). As serine can also be phosphorylated by Thr/Ser kinases, the conservation of this site indicates this is an important regulatory site. The NH_4_^+^-induced phosphorylation modulating NH_4_^+^ uptake seems to be a general mechanism to protect against NH_4_^+^ toxicity. Feedback inhibition of some AMTs is mediated by CALCINEURIN B-LIKE PROTEIN INTERACTING PROTEIN KINASE 15 (CIPK15) [39] and this may be the case in legumes. The pea genome does not seem to have any close relatives of AtCIPK15, while *M. truncatula* does have several (see Appendix A).

## 3. Assimilatory Steps and Enzymes

### 3.1. NIA, NITR2, and NiR Families

After uptake across the plasma membrane, NO_3_^−^ assimilation in plant cells begins in the cytosol with nitrate reductases (NIAs or NRs), which reduce NO_3_^−^ to nitrite (NO_2_^−^) [40,41], which is then transported into the chloroplasts by the HPP family transporters NITR2 [42] and reduced to NH_4_^+^ by nitrite reductases (NiRs) [43]. As in *A. thaliana,* two NIA and one NIR genes were found in the pea genome (see Appendix A). An important known regulatory phosphorylation site in NIAs is conserved in both PsNIAs (Figure 3A). However, no NITR2 gene could be detected in the pea genome. To verify this result, the AtNITR2 gene sequence was used to search other legume plant genomes, and one copy was found in each genome of soybean, *M. truncatula* and *L. japonicus*, indicating that the PsNITR2 may localise to a yet non-assembled part of the pea genome or missed annotation. Using the genomic sequence from *M. truncatula* to search the pea genome, a highly similar sequence was found on chromosome Ch5LG, then the gene structure was predicted by Genescan (http://hollywood.mit.edu/GENSCAN.html (accessed on 15 November 2021) and PsNITR2 was annotated (see Appendix A). 

### 3.2. Glutamine Synthetases (GS1 and GS2)

Glutamine synthetase (GS) catalyses the important step that combines N together with carbon (C) and the GS enzyme is involved in the reassimilation of NH_4_^+^ released from metabolic pathways, as well as assimilation of NH_4_^+^ sourced from the soil and produced by N-fixation. The enzyme sits at the crossroads between N and C assimilatory pathways and in cereal species and *A. thaliana*, the GS genes often map to QTLs that are important for nitrogen use efficiency [44]. There are cytosolic (GS1) and plastidic (GS2) forms of the enzyme and a seed specific GS was identified in *M. truncatula*, reviewed by Seabra and Carvalho [45].

Four glutamine synthetases I (GS1) proteins were found in the pea genome compared with five identified in *A. thaliana* (see Appendix A). Only one GS2 was detected in the pea genome like *A. thaliana*, while two were found in *M. truncatula* (see Appendix A). The post-translational regulatory residue S97 was conserved in all four pea GS2 proteins (Figure 3B). One Fd-GOGAT and one NADH-GOGAT were also found in the pea genome (see Appendix A). 

## 4. Regulatory Components

Nitrate responses in plants are governed by transcription factors, the most important being NODULE INCEPTION (NIN)-LIKE PROTEIN (NLPs) and downstream events include post-translational regulation by CIPKs and links to calcium signalling. Here we cover pea orthologues of CIPK23, calcium-dependent protein kinase III (CPKIII) and cyclic nucleotide gated channel 15 (CNGC15) from *A. thaliana* [46,47].

### 4.1. NLPs

The first-studied member of the NLP family was Nodule Inception (NIN) which was found to play an essential role in nodulation in *L. japonicus*, controlling both infection and N-fixation [48,49,50,51]. Phylogenetic analysis of NLPs reveals three groups [52]. In *M. truncatula*, MtNLP1, which belongs to the same clade as NIN, controls NO_3_^−^-suppression of nodulation, and physically interacts with NIN to regulate nodulation [53]. MtNIN and MtNLP2, the latter which belongs to the second group, drive leghemoglobin expression in nodules, with NLP2 being found only in papilionoid legumes [54]. In *A. thaliana*, AtNLP8, from the same clade as MtNLP2, regulates seed germination in response to NO_3_^−^ availability [55]. In the last group AtNLP6 and AtNLP7 are master PNR regulators in *A. thaliana* [56,57], and LjNLP4 control NO_3_^−^ suppression of nodulation in *L. japonicus* [58,59]. NO_3_^−^ induces a calcium influx in root cells that is dependent on AtNRT1.1/NPF6.3 triggering downstream responses [60]. This occurs through disassociation of AtCNGC15 with AtNRT1.1 leading to calcium permeation into the plant cell [47]. The calcium burst activates AtCIPK23/CLB1/9 to phosphorylate AtNRT1.1 at T101 to decrease its NO_3_^−^ transport activity [7,46]. At the same time, calcium activates CPKs (AtCPK10, AtCPK30 and AtCPK32) to phosphorylate AtNLP7 at S205 leading to its accumulation in the nucleus [56,61]. Then AtNLP7 activates transcription of PNR genes to promote NO_3_^−^ uptake and assimilation [14]. In *L. japonicus,* it is activated by NO_3_^−^ and directly activates CLE-RS2 expression, which is a root-derived mobile peptide that negatively regulates nodulation [59]. The role of NLP1/4 proteins in inhibiting nodulation when sufficient NO_3_^−^ is available allows legumes to nodulate, which is relatively costly in terms of energy compared to NO_3_^−^ assimilation, only when it is needed (see Section 5 below). There are five NIN and NLPs detected from the pea genome, which cover all three phylogenetic groups (see Appendix A). Amino acid residue S205 is conserved in all PsNLP proteins (see Appendix A). Mutants for PsNIN have been characterized, which show essentially identical phenotypes to orthologous mutants in other legumes [62]. MtNLP1, LjNLP4 and MtNLP2 all have clear orthologues in pea indicating they may have a conserved function in nodulation in pea.

### 4.2. CIPK23 and CBLs

There are 26 annotated CIPKs in the *A. thaliana* genome and 38 in that of *M. truncatula*. To our surprise, we could find only six CIPKs in the pea genome, which are all clustered in one major subclade (see Appendix A). No CIPK protein could be found in the AtCIPK15 subclade, which controls AMT1 activity (see Section 2.5 above). However, one homolog of AtCIPK23 could be identified, PsCIPK1. CALCINEURIN B-LIKE (CBL) proteins, AtCBL1 and AtCBL9, interact with AtCIPK23 to regulate NRT1.1 phosphorylation. There are 12 CBLs in pea, 15 in *M. truncatula* and 10 in *A. thaliana*, and all proteins are clustered in the phylogenetic tree (Appendix A). These observations indicate a conserved mechanism for NRT1.1 regulation (see Section 2.1 above).

### 4.3. CPKIII

Subgroup III CPK kinases are activated by calcium and phosphorylate AtNLP7 and facilitate shuttling of the transcription factor into the nucleus in *A. thaliana*. There are six CPKIIIs in the pea genome (see Appendix A), five in *M. truncatula*, and eight in *A. thaliana*. However, these is only one NLP protein, PsNLP4, in the AtNLP7 and AtNLP6 subclade (see Appendix A). If CPKIII phosphorylation of PsNLP4 is conserved, CPKIII can be predicted to mediate NO_3_^−^ suppression of nodulation in pea.

### 4.4. CNGC15

In *A. thaliana*, AtNRT1.1 de-suppresses AtCNGC15 calcium transport activity by NO_3_^−^, and calcium influx into cell activates calcium related kinases to facilitate NO_3_^−^ signalling [47]. The mechanism of how NO_3_^−^ causes AtCNGC15 to disassociate from AtNRT1.1 is still not clear, but could occur through direct effects of NO_3_^−^ on AtCNGC15 protein confirmation. There are three AtCNGC15 paralogues in *M. truncatula*, MtCNGC15a, b and c, while pea contained two, PsCNGC15a and c (see Appendix A), which are closest to MtCNGC15a and c, respectively. The apparent lack of a PsCNGC15b counterpart may be caused by limitations of the current genome assembly and annotation. Either way, potentially four NRT1.1s, with varied substrate preference may interact with the two CNGC15s in pea genome, which could form eight different protein pairs, suggesting a more complex mechanism of NO_3_^−^ sensing and signalling in pea and legume plants compared with *A. thaliana*.

### 4.5. PP2C

Protein phosphatase 2C (PP2C) proteins regulate NO_3_^−^ transporters, such as NRT1.1 and NRT2, by dephosphorylation [63,64]. 89 PP2C proteins were detected in pea, while 94 in *M. trucatula* and 80 in *A. thaliana* [65,66] (Appendix A). Proteins from all three species are clustered interactively, indicating conserved regulation by PP2C proteins.

### 4.6. CLE/CEP

Small peptides can be secreted by plant cells and are important circulating signals that control development including root lateral production and nodulation. These peptides signals are known as CLAVATA3/Embryo Surrounding Region-Related (CLE) or C-terminalLY ENCODED *PEPTIDE* (CEP) signals and their circulating concentration changes with the N status of the plant [67]. CLE peptides are very numerous in plants including legumes with >30 families, but they are difficult to annotate, and in legume models their numbers have been accurately determined [62]. Four homologues have been identified to be induced by NO_3_^−^ or Rhizobium in pea, Pscam040153, Pscam040702, Pscam040984 and Pscam041632, like *M. truncatula* which also has four, MtCLE12, 13, 34, 35 and *L. japonicus* which has three, LjCLE-RS1 to 3 [68] (see Appendix A). The function of other CLPs or CEPs still needs to be tested. The CLE peptides are recognized by LRR-RLKs receptors, these include HYPERNODULATON ABERRANT ROOT1 (HAR1) in *L. japonicus* or SUPER NUMERIC NODULES (SUNN) in *M. truncatula* to inhibit nodulation [69,70].

### 4.7. MiRNAs

For systemic regulation of N acquisition, shoot signals can mediated by small peptides as well as microRNAs [71]. miR2111s directly target TML, which negatively regulate nodulation [72]. Using miR2111 sequences from *L. japonicus* and *M. truncatula* to blast pea genome, we did detect several miR2111s in pea genome. However, since miRNAs are not well-annotated, it is difficult to comment more specifically on the situation in pea using our current knowledge. 

## 5. Nodulation

Including legumes in crop rotations is globally important as their association with N-fixing bacteria decreases the need for the application of N fertilizers. However, legumes have a homeostatic mechanism which allows the plant to balance the high energy cost of N-fixation with its N requirements. The application of N fertilizer inhibits the formation of new nodules and has a rapid suppressive effect on N-fixation in active nodules [73]. When growing under low nutrient supply the benefits of entering a symbiotic relationship for the plant outweigh the energy costs of carbon consumption by the partner. This balance is achieved in part by regulation of nodule number by the well-established autoregulation of nodulation (AON) pathway. AON is a systemic root-shoot-root control system mediated by circulating signalling CLE peptides and miRNA [71,74] (see Section 4.6 and Section 4.7 above). Acting opposite of that is the CEP signalling pathway, which is active under low NO_3_^−^ and promotes nodulation [67,75,76]. Agricultural soils in counties with developed economies contain high levels of residual N which limits legume nodulation and N-fixation. How the host legume senses and signals its N supply to regulate nodulation is therefore of fundamental importance for greener agriculture using lower inputs of chemical fertilizer in both industrialized and poorer countries. Maintaining legume yields without the addition of chemical N fertilizer is highly relevant to supporting more sustainable agriculture practices that provide major economic and social impact. Many of the transporters, assimilatory enzymes and regulatory components of NO_3_^−^ metabolism are expressed in nodules or regulate nodulation. Their pea homologs are listed in Table 1 below.

## 6. Conclusions

This survey of the pea genome is focused on N transport, assimilation and signalling and has identified some interesting differences between legumes and non-legumes. This gene assignment and family information is very dependent on the quality of the pea sequence information, nonetheless a few specific conclusions can be summarised for pea.

Pea has lower numbers of NRT2-type nitrate transporters, implying less reliance on nitrate from soil.Most post-translational regulatory sites for phosphorylation and nitrosylation are conserved in pea, such as NRT1.1, AMT1, NIA, GS2 and NLP.Pea has fewer CIPKs.

While N-homeostasis in legumes evolved to suit their native environments, these set-points are unlikely to be ideal for agricultural settings, and so stand to be improved. Some important targets for NUE in pea include the genes regulating nodulation, as these are key for understanding the control and regulation of diazotrophic N fixation. Like other crop species, the C:N interface in primary assimilation is also likely to be important for NUE. Pea appears to store less vacuolar NO_3_^−^ than non-legume species and increasing this storage capacity may be a useful target for improving NUE, providing the crop with a reservoir of N to sustain short-term requirements for growth. In intercropping systems, for example pea with barley, there is evidence that NUE is improved relative to a single pea crop [77]. Like other legumes, pea should have less dependence on N uptake from soil, with an optional N supply fixed by nodules. The suppression of nodulation by high levels of soil nutrients may have led to lower NUE in the non-nodulated plants, but more work is required to compare this trait in legumes in general.

## Figures and Tables

**Figure 1 genes-13-00158-f001:**
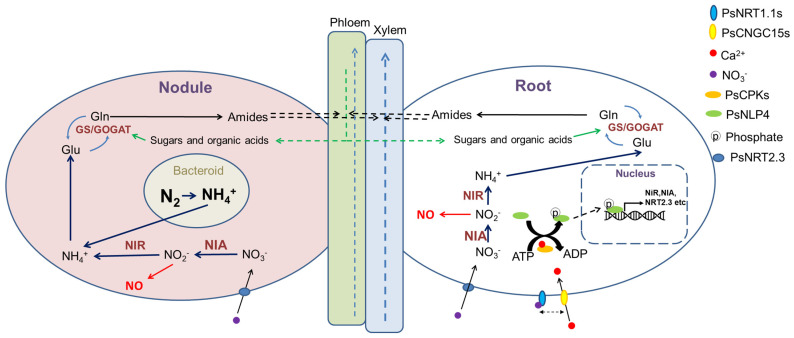
Diagram showing N transportation, fixation and assimilation in Pea. Figure redrawn from Murray et al., 2017 [5] Note pea is specifically an amide exporter, but other types of legume can transport organic N in the phloem as ureides and/or amino acids.

**Figure 2 genes-13-00158-f002:**
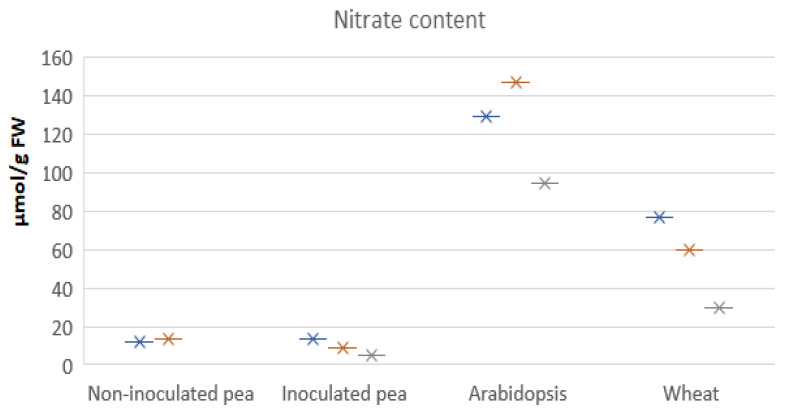
NO_3_^−^ concentrations in shoots of pea [25,26,27], *A. thaliana* [11,16,28] and wheat [29,30,31]. The tissue NO_3_^−^ concentrations, plotted using star symbols, were collected and pooled from these publications (Appendix A). FW: fresh weight. Blue, orange and grey stars indicate data from different publications.

**Figure 3 genes-13-00158-f003:**
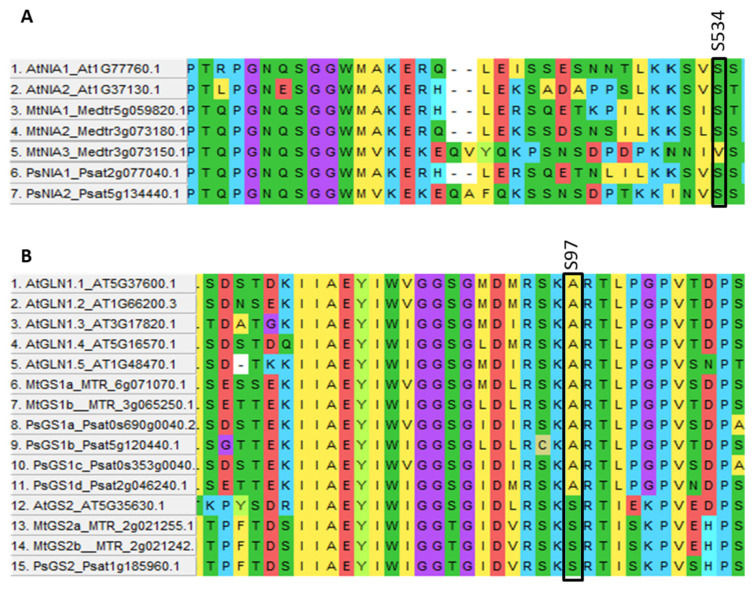
Phosphorylation site variation in NIA (**A**) and GS (**B**) proteins at S534 and S97 (**boxed**).

**Table 1 genes-13-00158-t001:** Pea genes involved in nodulation and NO_3_^−^ regulation.

Name	Gene Abbrev.	Accession Number
*NLPs*	*PsSYM35/PsNIN*	*Psat2g001120.1*
	*PsNLP1*	*Psat0s498g0160.3*
	*PsNLP2*	*Psat7g208800.1*
	*PsNLP3*	*Psat5g291840.4*
	*PsNLP4*	*Psat6g197400.1*
*NRT2s*	*PsNRT2.1*	*Psat4g155600.1*
	*PsNRT2.2*	*Psat7g149120.1*
	*PsNRT2.3*	*Psat4g113000.1*
*CLE/CEP/SUNNs*	*Pscam040153*	
	*Pscam040702*	*Psat7g164920.1*
	*Pscam040984*	
	*Pscam041632*	
*HAR1/SUNN*	*SYM29*	*Psat7g183240.1*
*PLENTY/RDN*	*PsRDN1*	*Psat0s3806g0040.1*
	*PsRDN2*	*Psat4g145800.4*
	*PsRDN3*	*Psat6g028240.1*
*TML*	*PsTML*	*Psat3g176480.1*
*NIA*	*PsNIA1*	*Psat2g077040.1*
	*PsNIA2*	*Psat5g134440.1*
*NIR*	*PsNIR*	*Psat7g123960.1*
*NITR2*	*PsNITR2*	*chr4LG4-85657742..85658907*
*AtCIPK23*	*PsCIPK1*	*Psat1g020800.1*
*GS1*	*PsGS1a*	*Psat0s690g0040.2*
	*PsGS1b*	*Psat5g120440.1*
	*PsGS1c*	*Psat0s353g0040.1*
	*PsGS1d*	*Psat2g046240.1*
*GS2*	*PsGS2*	*Psat1g185960.1*
*GOGAT*	*PsGLU1*	*Psat3g078160.3*
	*PsGLT1*	*Psat6g037200.1*

PsNITR2 annotation was not included in the database of URGI (https://urgi.versailles.inrae.fr/ (accessed on 1 November 2021)). Pscam040153, Pscam040984 and Pscam041632 are reported by Mens, et al. [68].

## Data Availability

Protein sequenced used are listed in Appendix A.

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
