# Peer review of "Inorganic Nitrogen Transport and Assimilation in Pea (*Pisum sativum*)"

_genes, 2022, doi:10.3390/genes13010158_

Round 1
Reviewer 1 Report
The paper presents a genome-based analysis of nitrogen-related genes in the newly published Pea genome. The reviewers (? I was unclear whether this counted as research or review, as there is no methodology, but the site locations/locus of Ps genes are given - presumably by annotation, with two cases where a related gene sequence was used to look for unannotated homologues in the pea genome) present a comparative in silico analysis of the genes involved in N uptake and also N-fixation, focusing on transporters and NH4 assimilation genes.
My first question would be whether this is review or research - the authors need to decide and if research, there needs to be clear material and methods presented. If Review, then what is the basis of the review? Current genome annotation?
Secondly, the authors indulge in a degree speculation as to why there are different numbers detected. A number of points:
1) There is no evidence presented from other pathways as to how comprehensive the current pea genome is
2) the comparisons between Arabidopsis and legumes are a little difficult. In Arabidopsis, there are often gene duplications adjacent to each other. Often, these gene have since diverged, very often in terms of their temporal and tissue-specific expression patterns. It might be more informative to identify how many distinct loci containing the gene of interest exist in Arabidopsis and compare those to the ones found in legumes.
3) the authors suggest that differences in copy number between Arabidopsis for specific gene functions argues for a difference in importance of that function for the species being considered. This is highly speculative and is not supported in the manuscript by any evidence.
4) In the abstract, the authors suggest that because pea has large seed, there is less need to capture nitrogen. However, medicago seed are actually quite small, so their statement is unlikely to be true.
Overall, this is an interesting pilot study, but I'm unclear whether it is review or research and there is a great deal of speculation, given that there is no hard evidence really offered
Author Response
My first question would be whether this is review or research - the authors need to decide and if research, there needs to be clear material and methods presented. If Review, then what is the basis of the review? Current genome annotation?
Answer: This is a review there is no new data or methods, all the genes listed are based on current pea genome annotation, except PsNITR2 was not annotated, so we used the genomic position. We prefer this paper as a review to reconstruct N pathway of pea from uncovered pathways of other species.
Secondly, the authors indulge in a degree speculation as to why there are different numbers detected. A number of points:
1) There is no evidence presented from other pathways as to how comprehensive the current pea genome is
Answer: During our review of the pea genome, we found these N related genes of primary assimilation are conserved when compared with Medicago and non-legumes like Arabidopsis. There are some notable differences, and we point these out to the reader. For example, fewer NRT2s, only PsNRT2.1 and PsNRT2.2 and interestingly this was supported by low nitrate content in pea tissues shown in previous publications (Figure 2). This evidence led us to predict a less nitrate uptake and more reliability on N fixation. Another surprising point concerned the CIPKs, only six proteins were found in pea genome. The influence of this dramatic decrease in this kinase family needs further study. We feel the aim of a comprehensive review of new genome data is to point out exciting new opportunities.
2) the comparisons between Arabidopsis and legumes are a little difficult. In Arabidopsis, there are often gene duplications adjacent to each other. Often, these gene have since diverged, very often in terms of their temporal and tissue-specific expression patterns. It might be more informative to identify how many distinct loci containing the gene of interest exist in Arabidopsis and compare those to the ones found in legumes.
Answer: We totally agree. To solve this problem, we used Medicago to address pea genes. From our result, we found an increased number of NRT1.1 in legumes, as Medicago has three NRT1.1s (MtNRT1.1a, b and c) and pea has four NRT1s (PsNRT1.1a, b c and d), while Arabidopsis has only one. This indicates a more complex nitrate sensing system in legumes and indicates a relationship between N uptake and N fixation by nodules. For NRT2s, compared to Medicago, the NRT2 number in Medicago and pea are the same, but PsNRT2.1 and 2.2 are truncated with possible loss of function, this is a special character in pea. The number of AMT2s is increased in both the legumes Medicago and pea, relative to non-legumes like Arabidopsis. The oddest case is in CIPKs, all six annotated PsCIPKs are clustered in the same clade of five clades. We could not detect more PsCIPKs, and not sure its influence on N regulation.
3) the authors suggest that differences in copy number between Arabidopsis for specific gene functions argues for a difference in importance of that function for the species being considered. This is highly speculative and is not supported in the manuscript by any evidence.
Answer: For NRT2s, which takes the major role in nitrate uptake, the lost function of PsNRT2.1 and PsNRT2.2 was supported by low nitrate contents from publications (Figure 2 and Supplemental Table 1). The AMT2 family is also extended in legumes, and one example is that MtAMT2.3 is functional for root symbiosis (https://doi.org/10.1105/tpc.114.131144). This evidence supports out statements, but we would agree that it is circumstantial.
4) In the abstract, the authors suggest that because pea has large seed, there is less need to capture nitrogen. However, medicago seed are actually quite small, so their statement is unlikely to be true.
Answer: Sorry, the logic was not very clear and confusing. We have modifed the abstract text to make this clearer to the reader (see line 20). In the abstract, we are discussing two aspects of decreased number of NRT2 transporters. First, compared with non-legumes, such as Arabidopsis, rice etc., Medicago and pea have fewer - only three high affinity NRT2s. This may indicate a lower capacity to adapt to changes in soil nitrate supply throughout the lifetime of the legume plants, this fits with the idea that these plants need less nitrate uptake from the soil, which is related to the symbiont N-fixation in nodules. Second, looking at the detail of PsNRT2.1 and 2.2 sequence, we found they are likely to have lost some nitrate transport function, this indicates less capacity to take up soil nitrate in pea. However, it is also very risky for a young seedling to establish before functional nodules are developed. This led us to think about the large seed of pea with more N storage. The young seedling needs N as a nutrition to germinate. Two sources can supply N needs for this process before nodule formation and N fixation. First, N can be supplied by storage in seed cotyledons and/or nitrate can be taken up by young roots from soil. The nitrate uptake from soil is mainly performed by NRT2 transporters. The big seed of pea with a large amount of N storage reduces dependency on root uptake from soil, which is consistent with our observation that both PsNRT2.1 and 2.2 have lost function. As the Reviewer mentioned, the seed of Medicago is very small with less N storage, therefore there is likely to be more reliance during seed germination on root nitrate uptake from soil, which is consistent with the fact that all the three MtNRT2 transporters are full-length (11-12 TM domains) and likely to be functional in Medicago. Line 20 in abstract was modified to clarify that the big seed of pea may be explained by its limited capacity for nitrate uptake having only one functional NRT2 during seed germination.
Reviewer 2 Report
This an intersting and timely review.
Some informations/citations are missing:
- At least some SLAC/SLAH proteins are nitrate transporters
- lines 59-67: this is speculation. This part should rewritten.
- NPF members should be named NPF and not NRT1.1. This nomenclature has been accepted and used since several years (Leran et al. TiPS).
- Figure 2, this is not cencentration (amount/volume) but content (amount/weight)
- CBLs and PP2C should also be analyzed in pea because these are important component of nitrate transport and sensing (Leran et al; Science Signaling)
- Gautrat et al should be discussed cited
- Several papers on legume nitrate transport are not cited: Medicago by Limami's group and Lotus by Chiurazzi group
Author Response
This an intersting and timely review.
Some informations/citations are missing:
- At least some SLAC/SLAH proteins are nitrate transporters
Answer: a paragraph was added for SLAC/SLAH proteins, Line 195 to 201 and Supplemental Figure 18.
- lines 59-67: this is speculation. This part should rewritten.
Answer: the paragraph was rewritten. Lines 60 to 78.
- NPF members should be named NPF and not NRT1.1. This nomenclature has been accepted and used since several years (Leran et al. TiPS).
Answer: NPF is a formal name for the NPF family proteins. NPF is a big protein family with redundant functions. However, the nitrate transceptor function of AtNPF6.3 is special sub-clade within the whole NPF family, and takes a major role in sensing and regulating the nitrate signal. In this paper, we focus on AtNPF6.3 and its orthologues in other species. As the numbers of members in NPF6s are different in different species, and orthologues numbered differently in difference species, this results in confusion. On another hand, MtNRT1.1s already got their names as MTNRT1.1a, b and c (see paper cited). Use of NRT1.1 can reduce the confusion between NPF6 subfamily and AtNPF6.3 orthologues. As we have stated previously the NPF name is confusing (see Murray et al. J. Expt, Bot. 2017), the transporters can mediate the movement of many other molecules across membranes including hormones (e.g. auxin), amino acids, glucosinolates, di-carboxylic acids not only nitrate and peptides. We used both names to introduce the family in the review.
- Figure 2, this is not cencentration (amount/volume) but content (amount/weight)
Answer: Many thanks for the correction, we made the change - see line 184.
- CBLs and PP2C should also be analyzed in pea because these are important component of nitrate transport and sensing (Leran et al; Science Signaling)
Answer: CBLs are discussed together with CIPKs (lines 303 to 308). PP2C regulation of AtNPF6.3 and NRT2 are discussed in an independent paragraph on PP2C (Lines 330 to 335 and Supplemental Figure 18).
- Gautrat et al should be discussed cited
Answer: Gautrat’s review on N systemic signalling was cited and discussed (Lines 352 to 353 and Line 368).
- Several papers on legume nitrate transport are not cited: Medicago by Limami's group and Lotus by Chiurazzi group
Answer: Limani’s reviews on nitrate transporters in legumes (line 51) and Chiurazzi’s chapter on Nitrate Transport and Signalling were cited (line 57).
We thank the Reviewer for their helpful suggestions to improve our manuscript.
Reviewer 3 Report
In the current review article, the authors tried to explain the main players in N transport in pea. Authors describe the involvement of the NO3 (both NRT1 and NRT2 families) and NH4 transporters in the process of N assimilation in pea. Large number of players involved in this process are described and compared to these involved in the same process in Arabidopsis and Medicago. In 17 supplementary figures authors present the sequence alignments of genes involved in N assimilation in pea with these of Arabidopsis and Medicago. The role of nodule inception proteins in the signaling cascade and the repression of nodule formation is also discussed.
Unfortunately genes/proteins involved in N transport and assimilation in pea are aligned only with these of Arabidopsis thaliana and Medicago truncatula genes. Including other already annotated genomes (for example these of Lotus japonicus and/or Oryza sativa) would definitely improve the level of the manuscript. Something similar was done only in supplementary Figure 2.
As a large number of phylogenetic trees are presented authors should explain how these trees were drown and which software were applied?
Further, the legend of the two figures presented in the manuscript should be improved. In Figure 2 for me is not clear what the different symbols presents. Are the authors sure that in the different experiments plotted in Fig. 2 plants were grown in similar conditions? To my knowledge in reference “28” plants were grown on 10mM KNO3, in reference “29” at 5mM and in some other at 2mM KNO3 respectively. Probably authors should include an additional table where the growing conditions of the plants included in Fig. 2 are clearly presented as it is known that the NO3 supply clearly influence NO3 content in the shoot.
There some minor points that should be also considered.
Authors should check carefully all supplementary figures! There are plenty of errors in the legend of these figures!
All abbreviation should be explained in the text (for example NIA, NIR, GS, at row 63)
Row 68 should move to 58, at the end of Fig1 legend.
Row 93: there is no supplemental Fig 2C.
Row 129: there no supplemental Fig3A. In my copy I have only 3C with the wrong legend.
Row 168 should be not “at least” but “up to 1 order of magnitude”.
Row 169 Authors say that this is true for both leaves and roots but no results or reference for roots are presented here!
Row 170. Please include a correct reference here, as in reference 23 authors observed clear difference in NO3 accumulation in nodulated plants in respect to non nodulated!
Row 213. Supplemental Fig. 8a but there is no b, same for Supplementary Figure 11b (row 235)
Row 313. In Supplementary Fig 17 CLE genes of Lotus are not included, neither Pscam040984.
Row 345 The title of Table 1 should be corrected. Genes involved in NO3?
The final conclusion (row 368-370) for me is very general and unclear.
Author Response
Unfortunately genes/proteins involved in N transport and assimilation in pea are aligned only with these of Arabidopsis thaliana and Medicago truncatula genes. Including other already annotated genomes (for example these of Lotus japonicus and/or Oryza sativa) would definitely improve the level of the manuscript. Something similar was done only in supplementary Figure 2.
Answer: We originally planned to include Lotus japonicus in our analysis of the NPF family, but to our great surprise, we found five LjNRT1.1 proteins, which is one more than the published paper (Plant Cell Environ. 2012 Sep;35(9):1567-81. doi: 10.1111/j.1365-3040.2012.02510.x. Epub 2012 Apr 19.). It looks as though the genome assembly and annotation of Lotus japonicus were renewed and new genes were found, so information about N transporters requires reviewing and we felt it was not a good model reference genome. For monocots, we found that both A and B subgroup NRT1.1 proteins from rice and maize clustered together as an outgroup of dicots proteins. This indicates that the nitrate sensing and signalling pathways are involved before the evolutionary separation of monocots and dicots. As this paper is to review transporters and signalling in pea, we only used the model plant of dicot, Arabidopsis, and model plant of legume, Medicago truncatula. Comparison between pea and rice will result variation between monocots and dicots, that is not specific for pea. We have added a note to the Supplementary Figure legend 2.
As a large number of phylogenetic trees are presented authors should explain how these trees were drown and which software were applied?
Answer: One paragraph of method was added (line 407).
Further, the legend of the two figures presented in the manuscript should be improved. In Figure 2 for me is not clear what the different symbols presents. Are the authors sure that in the different experiments plotted in Fig. 2 plants were grown in similar conditions? To my knowledge in reference “28” plants were grown on 10mM KNO3, in reference “29” at 5mM and in some other at 2mM KNO3 respectively. Probably authors should include an additional table where the growing conditions of the plants included in Fig. 2 are clearly presented as it is known that the NO3 supply clearly influence NO3 content in the shoot.
Answer: This is a very good advice. The N levels are listed in Supplemental Table 1.
There some minor points that should be also considered.
Authors should check carefully all supplementary figures! There are plenty of errors in the legend of these figures!
Answer: Thanks for you carefully checking. All are corrected.
All abbreviation should be explained in the text (for example NIA, NIR, GS, at row 63)
Answer: All modified. Lines 61 to 78.
Row 68 should move to 58, at the end of Fig1 legend.
Answer: moved to Line 60.
Row 93: there is no supplemental Fig 2C.
Answer: Deleted Supplemental Fig 2C.
Row 129: there no supplemental Fig3A. In my copy I have only 3C with the wrong legend.
Answer: Corrected as Supplemental Fig 3.
Row 168 should be not “at least” but “up to 1 order of magnitude”.
Answer: Corrected as “up to”.
Row 169 Authors say that this is true for both leaves and roots but no results or reference for roots are presented here!
Answer: Our apologies only shoot nitrate data was shown. The text has been corrected to ‘shoot’. Very few publications reported pea root nitrate concentrations.
Row 170. Please include a correct reference here, as in reference 23 authors observed clear difference in NO3 accumulation in nodulated plants in respect to non nodulated!
Answer: Corrected. We cite reference 23 at row 170 as suggested. The paper compared nitrate contents in pea and barley (Figure 2, Front. Plant Sci., 23 December 2020 | https://doi.org/10.3389/fpls.2020.602065). As we did not find more support data from barley and this paper did not involve a non-inoculated control, we could not draw a conclusion from one paper.
Row 213. Supplemental Fig. 8a but there is no b, same for Supplementary Figure 11b (row 235)
Answer: Corrected as Supplemental Fig. 8 and Supplemental Fig. 11
Row 313. In Supplementary Fig 17 CLE genes of Lotus are not included, neither Pscam040984.
Answer: Pscam040984 sequence were not found in the database or a published paper. LjCLE-RS1 to 3 are included in Supplemental Figure 17.
Row 345 The title of Table 1 should be corrected. Genes involved in NO3?
Answer: corrected as “NO3 regulation”.
The final conclusion (row 368-370) for me is very general and unclear.
Answer: The conclusions are based on the results of our analysis. As most gene families in pea are conserved with the model legume Medicago. For some points, we identify where more research is needed to uncover the mechanism and confirm differences between pea, Medicago and Arabidopsis as a non-legume model. Therefore, the conclusion is as clear as our understanding. For this review, we survey all the N pathways in pea and try to obtain an indication of likely N regulation. We hope our findings will help pea research to identify more targets for breeding. Our aim was to write a review and not a research article.